# The Management of Muscle Invasive Bladder Cancer: State of the Art and Future Perspectives

**DOI:** 10.3390/cancers17244017

**Published:** 2025-12-17

**Authors:** Antonio Cigliola, Brigida Anna Maiorano, Doga Dengur, Valentina Tateo, Chiara Mercinelli, Michela Piacentini, Sara Inguglia, Carlo Messina, Andrea Necchi

**Affiliations:** 1Department of Medical Oncology, IRCCS Ospedale San Raffaele, 20132 Milan, Italymercinelli.chiara@hsr.it (C.M.);; 2Vita-Salute San Raffaele University, 20132 Milan, Italy; 3Lewisham and Greewich Trust, London SE13 6LH, UK; doga.dengur@nhs.net; 4UOC Oncologia ARNAS Civico Benfratelli di Cristina, 90133 Palermo, Italycarlo.messina@arnascivico.it (C.M.)

**Keywords:** muscle-invasive bladder cancer, MIBC, neoadjuvant, immune checkpoint inhibitors, radical cystectomy, bladder preservation, antibody–drug conjugates, perioperative therapy

## Abstract

Muscle-invasive bladder cancer is a challenging disease due to its aggressive nature and high risk of systemic dissemination. Traditional treatment paradigms centered on cisplatin-based chemotherapy and radical cystectomy are being redefined by emerging systemic therapies, including immune checkpoint inhibitors and antibody–drug conjugates. This review summarizes current standards and highlights promising future directions in perioperative management.

## 1. Introduction

Bladder cancer accounted for approximately 83,190 new cases and 16,840 deaths in the United States in 2022 [1]. Urothelial carcinoma (UC) is the predominant histologic subtype, with around 25% of patients presenting with muscle-invasive bladder cancer (MIBC) at the time of diagnosis [2,3]. Due to the high risk of systemic dissemination and cancer-related mortality, MIBC has traditionally been treated aggressively in eligible patients. Indeed, the standard of care (SOC) for MIBC has included cisplatin-based neoadjuvant chemotherapy (NACT) followed by radical cystectomy (RC) with bilateral pelvic lymph node dissection (PLND) for decades. This approach has demonstrated a 5% absolute improvement in overall survival (OS) at 5 years and a 16% relative reduction in the risk of death at 10 years compared to surgery alone [4]. In selected patients, a bladder-sparing trimodal therapy (TMT)—consisting of maximal transurethral resection (TURBT) followed by concurrent chemoradiation—has also emerged as a viable alternative and is now a category 1 recommendation in the National Comprehensive Cancer Network (NCCN) guidelines [5,6].

Adjuvant chemotherapy, typically platinum-based, used to be considered for patients with a relapse risk after upfront surgery. However, due to its modest efficacy and considerable toxicity, its use remains limited and continues to be a subject of debate [7].

Nonetheless, growing recognition of the limitations of current strategies—such as chemotherapy ineligibility, long-term toxicity, and suboptimal response rates—has fueled a paradigm shift in the perioperative treatment landscape. Emerging systemic therapies, including immune checkpoint inhibitors (ICIs), antibody–drug conjugates (ADCs), and targeted therapies, which are already approved for metastatic or locally advanced disease, are now being investigated earlier in the treatment course [8,9]. These agents offer the potential for improved efficacy and more favorable toxicity profiles, potentially expanding treatment options not only for cisplatin-ineligible patients but also for those who are eligible, to enable bladder-sparing approaches based on treatment response.

At the same time, the optimal timing and strategy for approaching the earlier stages, whether neoadjuvant, adjuvant, or a combination as a perioperative approach, remain open questions. In recent years, numerous clinical trials have been initiated to address these uncertainties. In this review, we summarize recent advances in perioperative treatment strategies for MIBC, highlighting emerging evidence and ongoing studies that could redefine future SOC.

## 2. Materials and Methods

A structured literature search was performed using PubMed, EMBASE, and Cochrane databases, supplemented by a review of abstracts from major international oncology conferences, including the American Society of Clinical Oncology (ASCO) and genitourinary ASCO (GU-ASCO), the European Society of Medical Oncology (ESMO), and the American and European Associations of Urology (AUA/EAU). The search strategy incorporated combinations of relevant keywords and MeSH terms, such as “muscle-invasive bladder cancer,” “MIBC”, “neoadjuvant chemotherapy,” “neoadjuvant immunotherapy,” “adjuvant chemotherapy”, “adjuvant immunotherapy”, “perioperative”, “antibody–drug conjugates”, “targeted therapies”, “radical cystectomy”.

We included peer-reviewed clinical trials with post-hoc analyses and updates, and conference abstracts published in English up to 23 October 2025. Reviews, letters, editorials, case reports, and opinion pieces were excluded. A total of 33 studies met the predefined inclusion criteria and were included in this review.

This is a narrative/descriptive review. Studies were selected based on relevance, and no formal PRISMA methodology, risk-of-bias assessment, or meta-analysis was conducted.

## 3. Results

### 3.1. Neoadjuvant Treatment in MIBC

The most established benefit of NACT in MIBC is the significant improvement in OS, particularly among patients achieving a pathological complete response (pCR). Evidence from meta-analyses indicates that cisplatin-based NACT regimens are associated with a relative reduction in the risk of death of approximately 13% (HR ~0.87) and an overall pCR rate of 28.6%. Achieving pCR has been consistently associated with improved long-term outcomes, including a relative risk for OS of 0.45 (95% CI 0.36–0.56) and for recurrence-free survival (RFS) of 0.19 (95% CI 0.09–0.39) [10]. These findings support cisplatin-based NACT as the standard of care in appropriately selected MIBC patients.

Additionally, the neoadjuvant setting provides access to matched pre- and post-treatment tumor samples, facilitating biomarker discovery, and allows for the monitoring of minimal residual disease (MRD) through urinary tumor DNA (utDNA) analysis. Nonetheless, this approach has limitations. Delaying definitive local treatment in non-responders may compromise oncologic outcomes in cisplatin-refractory patients. On the other side, systemic therapy may prove unnecessary in specific subgroups, such as patients with limited disease or those who are ctDNA-negative at baseline. Furthermore, as the NACT carries a particular burden of toxicity, the approach may be particularly challenging in patients with rapidly worsening symptoms or emerging clinical complications, resulting in quality-of-life deterioration that may contraindicate proceeding with RC. Moreover, the use of NACT has significant limitations, particularly in patients who are ineligible for cisplatin-based regimens due to renal or cardiac dysfunction, comorbidities, or poor performance status. Furthermore, even among treated patients, recurrence and progression remain frequent, highlighting the need for more effective systemic strategies in the neoadjuvant setting [4].

In response to these challenges, several alternative neoadjuvant approaches have been investigated in recent years to develop more effective or less toxic treatments. Table 1 provides an overview of the most relevant clinical trials in the neoadjuvant setting, encompassing both completed studies and ongoing trials with preliminary published data. In the table, among published studies, PURE-01 and ABACUS provide the most robust and practice-informing evidence for single-agent immunotherapy, while combination regimens such as BLASST-1, AURA, and EV-103 suggest potentially greater efficacy but require further validation. Emerging ADC-based strategies remain promising but preliminary. Taken together, the data highlight an expanding therapeutic landscape, with ICIs representing the most clinically impactful addition to current neoadjuvant management.

#### 3.1.1. Chemo-Immunotherapy Combinations

Several early-phase trials have evaluated the feasibility and efficacy of combining ICIs with standard cisplatin-based chemotherapy as neoadjuvant treatment for MIBC to achieve more frequent and durable responses. These studies have reported encouraging rates of tumor downstaging and pCR, which may correlate with improved long-term outcomes.

The LCCC1520 phase II trial treated 39 patients with split-dose gemcitabine–cisplatin (GC) plus pembrolizumab, reporting a pCR rate of 36% and downstaging to <pT2N0 in 56% of patients, surpassing historical controls [11,12]. The HCRN GU14-188 trial similarly assessed GC plus pembrolizumab in two cohorts: among 43 cisplatin-eligible patients, pCR was achieved in 44% and overall pathologic response in 61%; in a second cohort of 37 cisplatin-ineligible patients receiving gemcitabine plus pembrolizumab, pCR and pathological response rate (pRR) were comparably high (45% and 52%, respectively) [13]. The BLASST-1 trial explored the addition of nivolumab to GC in 41 MIBC patients, achieving a pRR of 66% and pCR of 34% pCR, with a 12-month RFS rate of 85.4% [14]. A separate trial assessing neoadjuvant atezolizumab combined with GC found a pRR of 69%, pCR of 41%, and a 12-month RFS of 87% [15]. The phase 2 AURA trial assessed neoadjuvant avelumab-based regimens in 137 patients with MIBC undergoing RC. In the cisplatin-eligible cohort, pCR rates were 58% and 53% for ddMVAC-avelumab and GC-avelumab, respectively, with 36-month OS rates of 87% and 67%. In cisplatin-ineligible patients, avelumab monotherapy achieved a pCR rate of 32%, whereas the addition of paclitaxel-gemcitabine yielded only a 14% rate. The OS was similar between arms [16].

#### 3.1.2. Beyond Cisplatin: Immunotherapy, ADCs, and Targeted Therapies in the Neoadjuvant Setting for MIBC

For patients with MIBC who are ineligible for or decline cisplatin-based chemotherapy—representing around half of the MIBC population—RC without NACT used to represent the SOC. Moreover, as cisplatin therapy is associated with considerable toxicity even in eligible patients, there is a growing need to improve its outcomes through novel therapeutic strategies [4]. Increasingly, the goal is to achieve effective bladder-sparing approaches guided by response to treatment. Given this significant unmet clinical need, several studies have investigated alternative neoadjuvant strategies, including ICIs, ADCs, and targeted therapies, either as monotherapies or in combination.

The PURE-01 trial, the most extensive study to date in this setting, enrolled 143 patients (both cisplatin-eligible and -ineligible) to receive three cycles of pembrolizumab before RC. The study demonstrated a pCR rate of 39%, a pathologic downstaging rate of 55%, and 12- and 24-month event-free survival (EFS) rates of 84% and 72%, respectively, with manageable toxicity (6% grade 3–4 treatment-related adverse events [TRAEs]) [17]. Similarly, the ABACUS trial evaluated two cycles of atezolizumab in 95 patients, achieving a pCR rate of 31% and a 12-month RFS rate of 75% [18]. The umbrella OPTIMUS study evaluated 3 cycles of neoadjuvant retifanlimab monotherapy in cisplatin-ineligible patients. Among 20 treated patients, a 40% pCR rate and a 50% pRR were observed, with a manageable safety profile [19].

Combination immunotherapy strategies have also shown promise. Indeed, in addition to its cytotoxic effects, chemotherapy may also exert immunomodulatory activity by promoting immunogenic cell death, enhancing tumor antigen presentation, and altering the tumor microenvironment to promote promotes immune infiltration. These effects have provided the biological rationale for combining chemotherapy with ICIs, aiming to potentiate anti-tumor immune responses and improve pathologic response rates in the neo-adjuvant setting [20,21].

The NABUCCO trial investigated the dual checkpoint inhibition with ipilimumab and nivolumab in 24 patients, reporting a pCR rate of 46% and a pRR of 58%, suggesting potential synergy between CTLA-4 and PD-1 blockade [22]. The DUTRENEO trial prospectively stratified patients using a tumor inflammatory signature (TIS) and randomized those with “hot” tumors to receive durvalumab plus tremelimumab versus standard cisplatin-based chemotherapy [23]. Comparable pCR rates were reported between arms (34.8% vs. 36.4%), though stratification based on TIS did not clearly predict benefit from immunotherapy. The same combination was used in the recently published IMMUNOPRESERVE trial, where durvalumab and tremelimumab were combined with concurrent radiotherapy in an optic of bladder sparing approach, achieving a complete response in 81% of patients, with 2-year bladder-intact disease-free survival, distant metastasis-free survival, and OS rates of 65%, 83%, and 84%, respectively [24]. The PrECOG PrE0807 phase 1b trial evaluated neoadjuvant nivolumab with or without lirilumab, a monoclonal antibody targeting KIR receptors on Natural Killer cells, in cisplatin-ineligible patients. The treatment was well tolerated, with low rates of grade ≥3 TRAEs and no delays in surgery. Although pCR rates were modest (17–21%), 2-year RFS and OS rates exceeded 70%, supporting the feasibility of this approach [25]. More recently, the phase II GDFather-NEO trial introduced a novel strategy targeting Growth and Differentiation Factor 15 (GDF-15), a key mediator of resistance to PD-(L)1 blockade. In this study, visugromab (anti–GDF-15) combined with nivolumab achieved markedly improved outcomes compared with nivolumab monotherapy in cisplatin-ineligible or -refusing MIBC patients. The combination yielded pCR and MPR rates of 33.3% and 66.7%, respectively, versus 7.7% and 23.1% with nivolumab alone, and an ORR of 60% versus 15.4%. Notably, the combination regimen was well tolerated, with favorable safety and no unexpected toxicities [26].

ADCs also represent a promising class of agents in this setting. These compounds combine the specificity of monoclonal antibodies with the cytotoxic power of chemotherapy through the targeted delivery of potent cytotoxic payloads to tumor cells. The most well-known ADC in UC is enfortumab vedotin (EV), which targets Nectin-4, an adhesion molecule highly expressed in urothelial tumors, and delivers the microtubule-disrupting agent monomethyl auristatin E. This ADC has already entered routine clinical use in the metastatic setting, both as monotherapy and in combination with pembrolizumab [8,27,28].

In cohort H of the EV-103 trial (NCT03288545), 22 cisplatin-ineligible patients with cT2–T4aN0M0 MIBC received 3 cycles of neoadjuvant EV before RC. The study reported a pCR rate of 36.4% and a pathological downstaging (pDS) rate of 50.0%, with a favorable safety profile and no delays to surgery [29].

At ASCO 2025, the updated results of SURE-01 trial with neoadjuvant sacituzumab govitecan (SG), a Trop-2-directed ADC were presented. Among 33 efficacy-evaluable patients, the ypT0N0-x and ypT < 1N0-x rates were 36.4% and 39.4%, respectively, with a median follow-up of 14 months. The 12-month event-free survival (EFS) was 78.8%, and RFS was 100% in patients achieving ypT < 1N0-x [30].

At ASCO 2025, results from a single-agent study of disitamab vedotin (DV), a humanized anti-HER2 ADC, in 18 patients with HER2-overexpressing MIBC showed a pCR rate of 41.2% and a pDS rate of 64.7%, with no ≥grade 3 TRAEs, further reinforcing its potential as a neoadjuvant agent [31].

Encouraged by the promising efficacy of ICIs and ADCs as monotherapies, combination approaches are now under investigation. The HOPE-03 study evaluated a neoadjuvant regimen of DV combined with tislelizumab, an anti-PD1, in 51 patients with HER2-positive MIBC. Preliminary results showed a cCR rate of 56.5% and a disease control rate of 91.3%. Among patients who underwent radiotherapy-based bladder-preserving strategies, 35 achieved a complete response [32].

Alongside these emerging neoadjuvant strategies aiming to move beyond cisplatin-based chemotherapy, several ongoing trials are expected to provide pivotal data. Regarding targeted therapies, the SOGUG-NEOWIN (NCT06511648) trial is currently investigating an FGFR inhibitor, erdafitinib, with or without cetrelimab in the neoadjuvant setting for patients with FGFR-altered MIBC—representing a promising avenue for molecularly defined subgroups.

Collectively, these studies reflect the rapidly evolving landscape of neoadjuvant treatment in MIBC beyond cisplatin. Immunotherapy, ADCs, and targeted therapies are demonstrating encouraging pathologic responses and manageable safety profiles, paving the way for personalized, cisplatin-free strategies. Nonetheless, these approaches remain investigational, and confirmation through large randomized trials with long-term follow-up is essential to establish their survival benefit and support integration into the SOC.

### 3.2. Adjuvant Treatment in MIBC

The role of adjuvant systemic therapy in MIBC remains debated and is generally individualized based on prior NACT, pathological risk features, and cisplatin eligibility. While observational studies and meta-analyses suggest a potential survival benefit for cisplatin-based adjuvant chemotherapy in selected patients—particularly those who did not receive neoadjuvant therapy—the benefit remains limited in those previously receiving NACT, and postoperative complications often preclude its use. In this context, adjuvant immunotherapy has emerged as a promising alternative for patients with high-risk disease. This approach offers several theoretical advantages: it allows for the selection of patients based on postoperative pathology, thereby targeting systemic therapy to those at the highest risk of recurrence, and enables clinicians to address local disease control first. However, there are notable limitations as well. Compliance with adjuvant therapy tends to be lower than in the neoadjuvant setting, due to postoperative recovery and complications, and systemic treatment is introduced later in the disease course—potentially missing an opportunity for earlier intervention.

The phase III CheckMate 274 trial demonstrated a significant improvement in disease-free survival (DFS) with adjuvant nivolumab compared to placebo in patients with high-risk MIBC, independent of nodal status, PD-L1 expression, or prior neoadjuvant chemotherapy, leading to FDA approval of nivolumab in this setting. With an extended median follow-up of 36.1 months, nivolumab showed consistent DFS benefit both in the ITT population (HR 0.71) and in the PD-L1 ≥1% subgroup (HR 0.52). Importantly, interim OS analysis revealed a survival advantage (HR 0.76 in the ITT; HR 0.56 in PD-L1 ≥1%), and benefit was confirmed across additional endpoints such as nonurothelial tract recurrence-free and distant metastasis-free survival. Exploratory analyses in patients with MIBC reinforced efficacy regardless of PD-L1 status consolidating adjuvant nivolumab as a standard of care [33].

In contrast, the IMvigor010 trial did not meet its primary endpoint of DFS with adjuvant atezolizumab versus observation. However, post hoc analyses highlighted the predictive role of circulating tumor DNA (ctDNA). Among patients who were ctDNA-positive at baseline, atezolizumab significantly improved OS versus observation (HR 0.59), whereas ctDNA negativity was associated with favorable prognosis regardless of treatment. Moreover, dynamic ctDNA clearance during therapy correlated with prolonged survival, suggesting ctDNA as both a prognostic and predictive biomarker. These findings have shaped the design of the ongoing IMvigor011 trial, which selectively enrolls ctDNA-positive patients to validate biomarker-guided adjuvant immunotherapy [34]. In the recently reported results, adjuvant atezolizumab significantly improved both DFS (median 9.9 vs. 4.8 months; HR 0.64, *p* = 0.005) and OS (median 32.8 vs. 21.1 months; HR 0.59, *p* = 0.01) compared with placebo in ctDNA-positive patients after cystectomy. Importantly, patients who remained ctDNA-negative during surveillance achieved excellent outcomes without systemic therapy (DFS 95% at 1 year and 88% at 2 years), confirming the clinical utility of ctDNA-guided treatment selection in identifying those most likely to benefit from adjuvant immunotherapy while sparing others from unnecessary exposure [35].

Further supporting the strategy of ICIs, the phase III AMBASSADOR trial compared adjuvant pembrolizumab to observation in 702 patients with high-risk MIBC post-cystectomy. With a median follow-up of nearly 45 months, pembrolizumab significantly prolonged DFS (median 29.6 vs. 14.2 months; HR 0.73, *p* = 0.003). Although OS results are not yet mature, the DFS benefit was consistent across subgroups, reinforcing the role of ICIs in the adjuvant setting, although with a higher incidence of grade ≥3 adverse events compared to observation [36].

Looking ahead, strategies aimed at enhancing the efficacy of immunotherapy are under active investigation. One such approach involves the use of personalized mRNA-based cancer vaccines in combination with ICIs. While this has not yet been explored in the UC, trials like V940 (evaluating mRNA-4157 plus pembrolizumab in NSCLC) provide a rationale for applying this strategy in MIBC, particularly in patients with high-risk molecular profiles or minimal residual disease, potentially opening new avenues for personalized adjuvant treatment. An overview of the key clinical trials investigating adjuvant systemic therapy in MIBC is provided in Table 2, where CheckMate 274 and AMBASSADOR confirm the efficacy of nivolumab and pembrolizumab, while IMvigor011 highlights ctDNA as a biomarker to identify patients most likely to benefit, supporting a personalized adjuvant immunotherapy approach.

### 3.3. Perioperative Strategies

A perioperative treatment strategy that includes neoadjuvant therapy, RC, and subsequent adjuvant therapy is being increasingly explored in MIBC to maximize systemic disease control. This approach offers several potential advantages: it enables early initiation of systemic treatment, allows for postoperative risk stratification to guide adjuvant therapy (e.g., based on pathological findings or ctDNA status), and may provide synergistic effects—particularly in the context of immunotherapy. The strongest evidence supporting this approach comes from the phase III NIAGARA trial, which demonstrated that adding durvalumab to gemcitabine/cisplatin in the perioperative setting improved EFS (67.8% vs. 59.8%), OS (82.2% vs. 75.2%), and pCR rates (33.8% vs. 25.8%) compared with chemotherapy alone [37]. However, this strategy has its own limitations. Cumulative toxicity may increase, and postoperative complications often reduce eligibility or compliance with adjuvant therapy. In addition, neoadjuvant treatment may delay definitive surgery in non-responders, and overtreatment remains a concern in patients with low-risk disease or complete response [38]. Careful patient selection and biomarker-driven approaches will be key to optimizing outcomes in this setting.

#### Cisplatin-Ineligible MIBC Patients

In parallel, there is growing interest in perioperative regimens combining ADCs with ICIs, particularly for patients who are cisplatin-ineligible or who refuse chemotherapy.

The phase III KEYNOTE-905/EV-303 study (NCT03924895) evaluated perioperative EV plus pembrolizumab versus RC alone in patients with MIBC who were cisplatin-ineligible. As of the latest update presented at ESMO 2025, EV + pembrolizumab significantly improved EFS (median NR vs. 15.7 months; HR 0.40), OS (median NR vs. 41.7 months; HR 0.50), and pCR rate (57.1% vs. 8.6%; Δ 48.3%) compared with RC alone. Grade ≥3 AEs occurred in 71.3% of patients [39].

Further evidence from smaller phase II studies supports the feasibility of ADC–ICI combinations in this setting.

The NURE-Combo study investigated neoadjuvant nivolumab combined with nab-paclitaxel followed by RC and adjuvant nivolumab in patients ineligible for cisplatin or refusing it. Among 31 enrolled patients, the study met its primary endpoint, achieving a pCR of 32.3% and a major pathologic response (defined as ypT ≤ 1N0) rate of 70.9%. The 12-month EFS was 89.8% [40].

The phase II SURE-02 trial (NCT05535218) is evaluating neoadjuvant SG plus pembrolizumab, followed by adjuvant pembrolizumab, in patients with cT2–T4N0M0 MIBC. Notably, the protocol allows for bladder preservation in patients who achieve a stringent cCR, defined by negative MRI and absence of residual tumor on re-TURBT. In a pre-specified interim analysis presented at ASCO 2025, among 31 evaluable patients, the cCR rate was 38.7%, and the ypT ≤ 1N0-x rate was 51.6%. Grade ≥3 TRAEs occurred in 12.9% of patients, and SG was generally well tolerated. Molecular analyses revealed that luminal tumors were significantly more likely to achieve a pCR compared to non-luminal subtypes (73% vs. 25%, *p* = 0.04), whereas stromal-rich tumors were less responsive (*p* = 0.004). These findings suggest a promising role for perioperative SG plus pembrolizumab in enabling bladder preservation for a molecularly selected subset of patients.

Similarly, the EV-103 (NCT03288545) Cohort L study is investigating a perioperative approach using EV monotherapy in cisplatin-ineligible patients with previously untreated MIBC (cT2–T4aN0M0 or cT1–T4aN1M0). Patients received three cycles of neoadjuvant EV (1.25 mg/kg on Days 1 and 8 of each 3-week cycle), followed by RC, and six cycles of adjuvant EV. Among 51 treated patients, 82.4% completed surgery. The pCR rate was 34.0%, and the pDS rate was 42.0%, ≥grade 3 EV-related AEs occurred in 39.2% of patients [41].

Further evidence is expected from the large randomized phase III VOLGA (NCT04458311) trial which is evaluating durvalumab in combination with EV, with or without tremelimumab, as perioperative treatment in patients undergoing RC.

Concurrently, with the evolution of systemic therapies, novel intravesical drug delivery systems are being explored as part of bladder-sparing strategies. Initially developed for non–MIBC, these approaches are now being extended to MIBC. The phase II SunRISe-4 trial evaluated neoadjuvant TAR-200, a gemcitabine-releasing intravesical system, in combination with the anti–PD-1 antibody cetrelimab, versus cetrelimab alone, in patients ineligible for or refusing cisplatin-based chemotherapy. In the primary analysis (ESMO 2025), combination therapy achieved higher pCR (38% vs. 28%), pathologic overall response (pOR ≤ ypT1) (53% vs. 44%), and 1-year recurrence-free survival (77% vs. 64%) compared with cetrelimab monotherapy. Treatment was well tolerated, with no new safety signals and grade 3–4 adverse events below 10%. Biomarker analyses confirmed the potential of urinary tumor DNA (utDNA) and ctDNA as complementary tools for minimal residual disease (MRD) assessment. utDNA clearance and week-12 utDNA negativity significantly correlated with pCR, while ctDNA negativity was associated with prolonged recurrence-free survival, independent of treatment arm. These findings highlight the role of molecular biomarkers in refining response assessment and guiding bladder-sparing strategies following neoadjuvant therapy [42].

In Table 3, summarizing perioperative clinical trials in MIBC, the perioperative trial NIAGARA provides the strongest evidence supporting this strategy in cisplatin-eligible patients, while studies such as SURE-02, EV-103, and SunRISe-4 demonstrate the feasibility and efficacy of ADC–ICI combinations and bladder-sparing approaches in cisplatin-ineligible populations.

## 4. Discussion

Adjuvant, neoadjuvant, and perioperative strategies have all emerged as promising approaches in the treatment of MIBC (Figure 1), each demonstrating encouraging oncologic outcomes. Data from recent clinical trials have provided valuable insights that are actively shaping the future of disease management. Further important insights are expected from ongoing clinical trials across these settings, as summarized in Table 4, highlighting the ongoing efforts to develop more effective and less invasive therapeutic strategies.

In the neoadjuvant setting, the addition of ICIs to traditional chemotherapy has already shown encouraging results in terms of pCR and OS [11,14,15]. However, these results must be contextualized: patient populations in these trials often included fitter individuals eligible for cisplatin, which may limit generalizability to older or comorbid patients. Phase II studies evaluating immunotherapy monotherapy suggest comparable—or potentially superior—efficacy and safety relative to chemotherapy [17], yet their limited sample size and early-phase design constrain definitive conclusions. These findings point to a paradigm shift in the neoadjuvant management of MIBC. The evidence is even more substantial in the adjuvant setting, where the results of the CheckMate 274 [33] and AMBASSADOR [36] trials are reshaping clinical practice by offering a practical and better-tolerated alternative to chemotherapy. These studies mark a move away from poorly tolerated platinum-based regimens towards more patient-friendly and efficacious immunotherapeutic options.

The NIAGARA trial represents a pivotal advancement in the perioperative treatment landscape with potential practice-changing implications [37]. Several ongoing studies are exploring the integration of ADCs and immunotherapy, raising the question of whether standard chemotherapy might eventually be excluded altogether from perioperative strategies. Moreover, novel systemic therapies—including ICIs, ADCs, and targeted agents—are beginning to challenge cisplatin-based chemotherapy, particularly in selected patient populations, by offering improved efficacy and safety profiles. However, robust comparative data are still lacking, and long-term safety and survival outcomes remain under evaluation.

While the abundance of trials and emerging data has not yet produced a definitive treatment algorithm, one key takeaway is the recognition of pCR as a robust surrogate marker for OS. In parallel, the use of ctDNA to detect MRD and guide risk-adapted treatment strategies has introduced a new dimension to personalized care. Robust evidence shows that ctDNA positivity—both before and after radical cystectomy—is strongly prognostic for recurrence and survival, with detection sensitivities exceeding 90% in some studies [43]. Furthermore, ctDNA dynamics during neoadjuvant or perioperative therapy can provide early indications of treatment response [44], identify patients unlikely to benefit from further systemic therapy, and potentially support treatment de-escalation in those achieving molecular clearance. Incorporating additional biomarkers, such as ctDNA methylation profiles and other genomic or epigenomic signatures, could further refine predictive accuracy, enabling highly tailored perioperative strategies and potentially reducing overtreatment. The practical integration of these biomarkers in clinical practice, however, faces several challenges, including variability in test availability, standardization issues, turnaround time, and inter-laboratory differences. Further prospective clinical studies are required to validate biomarker-driven strategies and confirm their safety and efficacy before they can be routinely applied in patient care. In this context, broader adoption of ctDNA and other molecular biomarkers will also require harmonised assay methodologies and stronger evidence of clinical utility, while practical constraints, particularly cost and access to advanced testing, remain important barriers to routine implementation.

Despite recent advances, several important questions remain unanswered. One key issue concerns the prognostic implications of achieving a pCR with immunotherapy alone compared to pCR achieved through a combination of immunotherapy and chemotherapy. While both approaches may lead to complete tumor regression, it remains unclear whether the depth or durability of response differs between the two modalities. This distinction could have significant implications for tailoring treatment intensity and selecting the most appropriate therapeutic strategy for individual patients. Another critical question is whether a pCR following neoadjuvant immunotherapy alone may be sufficient to justify the omission of adjuvant therapy. If a complete response before surgery proves to be a reliable surrogate marker for long-term disease control, it could potentially spare patients from additional systemic treatment and its associated toxicity. However, current evidence is limited, and long-term data are needed to determine the safety and efficacy of this de-escalation approach. Indeed, none of the trials conducted so far have been designed to separate the role of neoadjuvant from that of adjuvant therapy on EFS and OS outcomes in this setting. Furthermore, no reliable prognostic biomarkers have been identified to date.

These considerations must be balanced against additional dimensions of treatment success that extend beyond oncologic efficacy. Notably, toxicity profiles, perioperative safety, patient-reported outcomes, and functional results are especially relevant for older or frail patients, who constitute a substantial proportion of the MIBC population and may experience disproportionate treatment-related morbidity. A more comprehensive evaluation of these aspects is essential to fully assess the feasibility and real-world applicability of multimodal perioperative approaches, ensuring that therapeutic intensification does not come at the expense of tolerability, functional preservation, or quality of life. Finally, the role of organ-sparing strategies—such as TURBT or chemoradiotherapy—in enhancing the efficacy of perioperative immunotherapy remains to be defined. These less invasive approaches may offer significant advantages in terms of quality of life, particularly for selected patients with favorable clinical or molecular features. Nevertheless, their integration with immunotherapy in the perioperative setting requires further investigation to determine the optimal sequencing, patient selection, and impact on oncologic outcomes. Emerging ctDNA data and the high pCR rates achieved with modern neoadjuvant and perioperative regimens, particularly ICI-based and ADC–immunotherapy combinations, support the growing interest in bladder preservation. Nevertheless, early bladder-sparing experiences still report non-negligible rates of local and distant relapse, calling for prudence.

These questions emerge in a context where the very definition of treatment success is evolving. Historically, the goal was to cure the disease—an outcome that can be achieved with neoadjuvant, adjuvant, or perioperative strategies. However, today’s treatment goals increasingly emphasize bladder preservation, reflecting both physician and patient priorities. Importantly, bladder preservation implies avoiding not only RC but also radiotherapy, which remains central to trimodal therapy but often leads to significant bladder dysfunction. This shift naturally deprioritizes adjuvant strategies, which inherently require RC. In contrast, neoadjuvant and perioperative approaches are more aligned with future trends, as they offer greater potential for bladder-sparing treatment.

This evolving landscape highlights the urgent need for a validated, objective definition of clinical response—ideally one that approximates pCR and supports decision-making in bladder-preserving protocols. While pathological response remains the most reliable predictor of survival, a non-invasive surrogate would facilitate broader implementation of bladder-sparing approaches. In this context, the phase II study by Galsky et al. offers encouraging evidence and points toward a novel approach—not only in terms of defining cCR, based on negative biopsy, urine cytology, and imaging—but also in recognizing its value for safely forgoing immediate cystectomy, with a positive predictive value of 0.97 for favorable outcomes, highlighting the potential for effective bladder-sparing strategies guided by non-invasive assessment [45].

This review has limitations not only related to its methodology but also inherent to the studies considered. Most of the available evidence comes from early-phase trials or conference abstracts, overall survival data are lacking for many regimens, and the selection of patients often favors fit, cisplatin-eligible individuals, which may limit generalizability. In addition, heterogeneity in trial design, endpoints, and biomarker assessments further complicates interpretation. Acknowledging these limitations provides important context, offering an updated overview of the current landscape and suggestions for future possibilities.

## 5. Conclusions

The management of MIBC is undergoing a fundamental transformation, driven by the integration of ICIs, ADCs and target therapies, biomarker-guided strategies, and evolving treatment goals that increasingly prioritize organ preservation and patient quality of life. While traditional cisplatin-based chemotherapy remains a cornerstone in selected patients, emerging data from neoadjuvant, adjuvant, and perioperative trials underscore the growing role of ICIs and novel agents. However, several key challenges—such as optimizing patient selection, defining reliable surrogate endpoints, and determining the safest de-escalation strategies—must still be addressed to fully realize the promise of bladder preservation and the associated improvement in quality of life.

## 6. Summary Section and Future Perspectives

The evolving treatment paradigm for MIBC emphasizes a shift towards immunotherapy-based neoadjuvant and perioperative strategies, integration of novel systemic agents like ADCs, ctDNA-guided personalization, and selective bladder preservation. Emerging data on ctDNA, together with promising results from ADC-immunotherapy combinations, lay the groundwork for potentially surpassing radical cystectomy in the future. These advances may redefine clinical decision-making, offering improved survival, reduced toxicity, and enhanced quality of life, while guiding future research towards optimal sequencing, biomarker-driven selection, and bladder-sparing strategies.

## Figures and Tables

**Figure 1 cancers-17-04017-f001:**
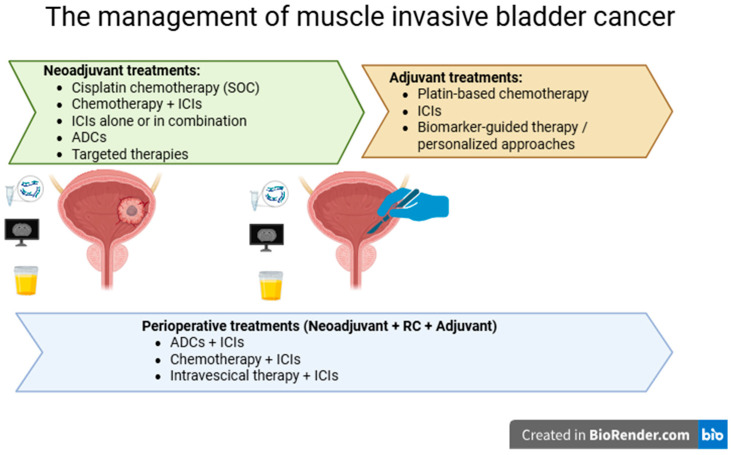
Current and emerging treatment strategies for the management of muscle-invasive bladder cancer.

**Table 1 cancers-17-04017-t001:** Neoadjuvant Clinical Trials in Muscle-Invasive Bladder Cancer with Published or Preliminary Results.

Study Name	Phase	Target Population (Number of pts)	Administered Drugs	Primary Endpoint	Results
PURE-01	II	MIBC, cisplatin-eligible & ineligible (143)	Pembrolizumab	Pathologic complete response (pCR)	pCR 39%, downstaging 55%, 12-mo EFS 84%, manageable toxicity
ABACUS	II	MIBC, cisplatin-ineligible (95)	Atezolizumab	pCR	pCR 31%, 12-mo RFS 75%
LCCC1520	II	MIBC, cisplatin-eligible (39)	Gemcitabine/Cisplatin split-dose + Pembrolizumab	pCR	pCR 36%, downstaging <pT2 56%
HCRN GU14-188	Ib/II	MIBC, cisplatin-eligible (43) & ineligible (37)	Cisplatin cohort: GC + Pembrolizumab; Cisplatin-ineligible: Gemcitabine + Pembrolizumab	pCR	pCR 44% (eligible), 45% (ineligible), overall pathologic response 61% & 52%
BLASST-1	II	MIBC (41)	GC + Nivolumab	Pathologic response rate (pRR)	pRR 66%, pCR 34%, 12-mo RFS 85.4%
AURA	II	MIBC, cisplatin-eligible & ineligible (137)	ddMVAC-avelumab, GC-avelumab, avelumab ± paclitaxel-gemcitabine	pCR	Cohort eligible: pCR 53–58%, 36-mo OS 67–87%; Ineligible: pCR 14–32%, OS similar
NABUCCO	I/II	MIBC (24)	Ipilimumab + Nivolumab	pCR	pCR 46%, pRR 58%
DUTRENEO	II	MIBC, stratified by Tumor Inflammatory Signature (TIS)	Durvalumab + Tremelimumab vs. Cisplatin-based chemo	pCR	pCR 34.8% vs. 36.4%, TIS stratification did not clearly predict benefit
IMMUNOPRESERVE	II	MIBC (bladder-sparing)	Durvalumab + Tremelimumab + Concurrent Radiotherapy	Complete response (CR)	CR 81%, 2-yr bladder-intact DFS 65%, OS 84%
PrECOG PrE0807	Ib	MIBC, cisplatin-ineligible	Nivolumab ± Lirilumab	pCR	pCR 17–21%, 2-yr RFS & OS >70%
EV-103 Cohort H	Ib/II	MIBC, cisplatin-ineligible (22)	Enfortumab Vedotin, 3 cycles pre-RC	pCR	pCR 36.4%, pathological downstaging 50%, safe, no surgery delays
SURE-01	II	MIBC, cisplatin-ineligible (21)	Sacituzumab Govitecan	pCR	cCR 47.6%, pCR 37.5%, ypT ≤ 1N0-x 43.7%
ChiCTR2300068270	II	HER2+ MIBC (18)	Disitamab Vedotin	pCR	pCR 41.2%, pDS 64.7%, no ≥ grade 3 TRAEs
HOPE-03	II	HER2+ MIBC (51)	Disitamab Vedotin + Tislelizumab	cCR/Disease control rate	cCR 56.5%, disease control 91.3%, 35 bladder-preserving CR

**Table 2 cancers-17-04017-t002:** Adjuvant Clinical Trials in Muscle-Invasive Bladder Cancer with Published or Preliminary Results.

Study Name	Phase	Target Population (Number of pts)	Administered Drugs	Primary Endpoint	Results
CheckMate 274	III	High-risk MIBC post-RC (patients with or without prior NACT, 709)	Nivolumab vs. Placebo	Disease-Free Survival (DFS)	Median not reached vs. placebo; HR 0.71 (ITT) vs. placebo, HR 0.52 (PD-L1 ≥ 1%) vs. placebo
IMvigor010	III	High-risk MIBC post-RC (patients with or without prior NACT, 809)	Atezolizumab vs. Observation	DFS	No significant DFS benefit overall vs. observation; post hoc ctDNA+ subgroup HR 0.59 vs. observation
AMBASSADOR	III	High-risk MIBC post-RC (702)	Pembrolizumab vs. Observation	DFS	Median 29.6 vs. 14.2 mo; HR 0.73 vs. observation (*p* = 0.003); higher incidence of grade ≥3 AEs vs. observation
IMvigor011	III	ctDNA positive MIBC post-RC	Atezolizumab vs. Observation	DFS	Median 9.9 vs. 4.8 mo; HR 0.64 vs. observation (*p* = 0.005) in ctDNA+ patients

**Table 3 cancers-17-04017-t003:** Perioperative Clinical Trials in Muscle-Invasive Bladder Cancer with Published or Preliminary Results.

Study Name	Phase	Target Population (Number of pts)	Administered Drugs	Primary Endpoint	Results
NIAGARA	III	MIBC, cisplatin-eligible (number not specified in summary)	Gemcitabine/Cisplatin ± Durvalumab → RC → Adjuvant Durvalumab	Event-Free Survival (EFS), OS, pCR	EFS 67.8% vs. 59.8%, OS 82.2% vs. 75.2%; pCR 33.8% vs. 25.8% favoring durvalumab
NCT03558087	II	MIBC, cisplatin-eligible (76)	Gemcitabine/Cisplatin + Nivolumab with bladder-preserving strategy	Complete Clinical Response (cCR)/PPV for composite outcome	cCR 43%, PPV 0.97, 32 patients deferred RC
NURE-Combo	II	MIBC, cisplatin-ineligible or refusing cisplatin (31)	Nivolumab + Nab-Paclitaxel → RC + adjuvant Nivolumab	pCR	pCR 32.3%, major pathologic response 70.9%, 12-mo EFS 89.8%
SURE-02	II	MIBC, cisplatin-ineligible (31)	Sacituzumab Govitecan + Pembrolizumab → RC → Adjuvant Pembrolizumab	cCR/ypT ≤ 1N0-x	cCR 38.7%, ypT ≤ 1N0-x 51.6%, grade ≥3 TRAEs 12.9%
EV-103 Cohort L	Ib/II	MIBC, cisplatin-ineligible (51)	Enfortumab Vedotin neoadjuvant 3 cycles → RC + PLND → adjuvant 6 cycles EV	pCR/pDS	pCR 34%, pDS 42%, ≥grade 3 TEAEs 39.2%
GDFather-NEO	II	MIBC, cisplatin-ineligible or refusing cisplatin (31 enrolled, 28 efficacy-evaluable)	Visugromab (anti–GDF-15) + Nivolumab → RC or re-TURBT	pCR	pCR 33.3% vs. 7.7% with nivolumab + placebo; ORR 60% vs. 15.4%; N/V well tolerated with no unexpected toxicities.
SunRISe-4	II	MIBC, cisplatin-ineligible or refusing cisplatin (122 randomized, 120 treated)	TAR-200 (intravesical gemcitabine) + Cetrelimab → RC	pCR	pCR 42% vs. 23% with cetrelimab monotherapy, combination well tolerated, grade ≥3 AEs 11%, no treatment-related deaths

**Table 4 cancers-17-04017-t004:** Ongoing Clinical Trials of Neoadjuvant, Adjuvant, and Perioperative Strategies in Muscle-Invasive Bladder Cancer.

ClinicalTrials.gov ID (Name)	Phase	Drugs Used/Strategy
NEOADJUVANT		
NCT04730219 (SOGUG)	II	Erdafitinib ± Cetrelimab (FGFR-altered MIBC)
NCT05241340 (RAD-VACCINE)	II	Neoadjuvant Sasanlimab With Radiation
NCT05839119 (TASUC-Neo)	II	gemcitabine and cisplatin and If it has the testosterone receptor participants will receive a medication called Degarelix that lowers testosterone levels
NCT05581589	II	Sacituzumab Govitecan in Non-Urothelial Muscle Invasive Bladder Cancer
NCT06263153	II	Futibatinib in Combination With Durvalumab
ADJUVANT		
NCT06305767	II	Pembrolizumab alone or in combination with vaccine V940
NCT06682728	II	Sacituzumab Govitecan and Nivolumab
PERIOPERATIVE		
NCT03732677 (CA078)	III	Nivolumab + Gemcitabine/Cisplatin (perioperative)
NCT03924895 (KEYNOTE-866)	III	Pembrolizumab + Gemcitabine/Cisplatin (perioperative)
NCT04960709 (ENERGIZE)	III	GC ± Nivolumab ± Linrodostat
NCT04700124 (EV-304/KEYNOTE-B15)	III	Pembrolizumab + Enfortumab Vedotin vs. Chemo
NCT04458311 (VOLGA)	III	Durvalumab + Enfortumab Vedotin ± Tremelimumab
NCT05328336	II	Tislelizumab in Combination With Nab-Paclitaxel
NCT06133517	II	Sacituzumab govitecan + Zimberelimab (AB 122) + Domvanalimab (AB 154)

## Data Availability

No new data were created or analyzed in this study.

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
