# Peer review of "The Management of Muscle Invasive Bladder Cancer: State of the Art and Future Perspectives"

_cancers, 2025, doi:10.3390/cancers17244017_

Round 1

Reviewer 1 Report

Comments and Suggestions for Authors

This paper : The management of Muscle invasive bladder cancer: State of the 2 Art and Future Perspectives, is an update of current clinical protocols, with special emphasis on immunotherapy and cytotoxic antibodies administered in both neoadjuvant and adjuvant settings.

It is a well-structured paper with very appropriate conclusions regarding current results and future perspectives. The importance of suitable biomarkers is highlighted.

It would have been more interesting and engaging if a figure illustrating the mechanisms of action of the compounds and the monitoring markers had been included; however, this does not detract from the overall quality of the work presented.

Minor errors:

1- On line 149, it says "SoC," and I understand it should be all in capital letters, written as SOC.

2- The heading 3.3.1 says: "Cisplatin-eligible MIBC patients," and it should say "Cisplatin-ineligible MIBC patients."

Author Response

Response to Reviewer

We thank the Reviewer for the constructive and insightful comments on our manuscript “The Management of Muscle-Invasive Bladder Cancer: State of the Art and Future Perspectives.”

We appreciate the positive assessment regarding the structure, conclusions, and discussion of biomarkers.

Regarding the specific points raised:

  1. The acronym SoC has been corrected to SOC as suggested.
  2. The heading 3.3.1 has been amended from “Cisplatin-eligible MIBC patients” to “Cisplatin-ineligible MIBC patients,” in accordance with the Reviewer’s comment.

We are grateful for the helpful feedback, which has improved the clarity and accuracy of the manuscript.

Reviewer 2 Report

Comments and Suggestions for Authors

The manuscript is comprehensive and well-structured, with valuable updates on evolving neoadjuvant, adjuvant, and perioperative strategies. Minor enhancements-such as clearer synthesis of trial data, concise discussion of biomarkers, improved readability, and expanded context for bladder-preservation approaches-would further strengthen its clarity and clinical relevance. Some concerns need to be addressed for improvement of the paper.

  • The manuscript is dense in several sections, with lengthy descriptions of trial outcomes and overlapping details that may reduce readability. Streamline narrative flow by summarizing key findings, avoiding repetition, and using clearer transitions between neoadjuvant, adjuvant, and perioperative sections.
  • Manuscript provides extensive trial summaries, limited critical interpretation of results, limitations, and comparative insights. Enhance the discussion by highlighting strengths/weaknesses of major trials, differences in patient populations, and implications for clinical practice.
  • The manuscript reports numerous emerging therapies but lacks a unified perspective regarding how these evolving strategies redefine clinical decision-making. Add a concise summary section synthesizing the evolving treatment paradigm, emphasizing how new data may shift the standard of care.
  • Although ctDNA and molecular signatures are mentioned, the practical clinical integration and limitations of these biomarkers are not sufficiently detailed. Expand on the role of biomarkers in treatment selection, MRD assessment, and tailoring neoadjuvant/adjuvant therapies, including challenges in real-world application.
  • Tables summarizing clinical trials are comprehensive but dense, lacking contextual interpretation to help readers appreciate relative importance of evidence. Add explanatory text summarizing conclusions drawn from each table and emphasize which trials currently influence clinical practice.
  • Some sentences are long and complex, reducing clarity. Minor inconsistencies occur in terminology (e.g., pCR vs ypT0), abbreviations, and referencing style. Perform a thorough language and style edit, ensure uniform use of terms/abbreviations, and cross-check references for formatting accuracy.
  • Given the shifting emphasis toward organ preservation, the discussion on bladder-sparing approaches is relatively brief and could be expanded. Elaborate on emerging evidence for bladder-preserving protocols, criteria for selection, and future directions in integrating systemic therapies for bladder-sparing.

Author Response

Response to Reviewer

We thank the Reviewer for the thorough and constructive evaluation of our manuscript. We appreciate the positive feedback regarding its structure, comprehensiveness, and relevance to evolving neoadjuvant, adjuvant, and perioperative strategies in muscle-invasive bladder cancer.

In response to the Reviewer’s comments, the following revisions have been made:

Readability and narrative flow

Several dense sections have been streamlined to reduce redundancy and improve transitions between neoadjuvant, adjuvant, and perioperative discussions.

Revisions performed at lines 108–109, 179, 237, and 335–336.

Critical interpretation of clinical trials

The discussion has been expanded to provide deeper interpretation of major trials, including strengths/weaknesses, differences in enrolled populations, and implications for clinical practice.

Revisions performed at lines 417–419, 435–436, 450–458, 475–482, and 513–520.

Unified perspective on evolving therapeutic strategies

A concise summary section has been added to synthesize how new clinical data may reshape treatment paradigms and influence future standards of care.

Revisions performed at lines 532–541.

Role of biomarkers in clinical practice

The biomarker section has been expanded to better address clinical integration of ctDNA and molecular signatures, including limitations, MRD assessment, and implications for therapy tailoring.

Revisions performed at lines 450–458.

Contextual interpretation of tables

Explanatory text has been added for each clinical table to summarize conclusions and highlight trials with current clinical relevance.

Revisions performed for Table 1 at lines 125–131; Table 2 at lines 305–308; Table 3 at lines 382–386.

Language clarity, consistency, and style

The manuscript has undergone extensive language and terminology editing to correct long and complex sentences, ensure uniform use of abbreviations (e.g., pCR vs ypT0), and standardize referencing style.

Revisions performed at line 351 and in several additional sections of the manuscript.

Bladder-preservation approaches

The section on bladder-sparing strategies has been expanded, adding details on patient-selection criteria, emerging evidence, and integration of systemic therapies into organ-preservation protocols.

Revisions performed at lines 536–540.

We believe that these revisions substantially enhance the clarity, interpretative depth, and clinical applicability of the manuscript. We are grateful to the Reviewer for the constructive feedback that has significantly improved the quality of our work.

Reviewer 3 Report

Comments and Suggestions for Authors

This manuscript provides a timely and comprehensive overview of perioperative systemic therapy in muscle-invasive bladder cancer and summarises many of the pivotal and emerging studies shaping current practice. The topic is important, the structure is clear, and the authors have assembled a substantial body of data. However, several key elements needed for a rigorous scholarly review are missing, and the manuscript in its present form reads largely as a narrative listing of trials rather than a critically appraised synthesis.

The most substantive limitation lies in the methodology. Although the authors describe conducting a “systematic literature search,” no reproducible methods are provided: the search strategy, databases consulted, inclusion and exclusion criteria, screening procedure, and number of records retrieved and selected are not reported. No PRISMA-style flow diagram is included, and there is no assessment of study quality or risk of bias. Without these elements, the review cannot be considered systematic, and the Methods section should either be expanded substantially to meet basic standards or rewritten to reflect a narrative review approach with an accompanying acknowledgement of limitations.

The Results section is detailed but predominantly descriptive. The manuscript would benefit from clearer comparative synthesis across regimens, including explicit discussion of heterogeneity in patient populations, endpoints, and trial designs. Integrating these data into a more unified interpretation would substantially improve the utility of the review for clinicians. The addition of a simple treatment-algorithm figure or a more explicit clinical decision framework would further strengthen the manuscript’s practical value.

The Discussion is focused on efficacy outcomes, particularly pCR and DFS, but pays limited attention to toxicity, perioperative safety, patient-reported outcomes, functional outcomes after bladder preservation, or the specific challenges in older or frail patients who represent a significant proportion of the MIBC population. These aspects are essential when evaluating multimodal perioperative strategies and should be addressed more fully.

The “future perspectives” section, while highlighting key ongoing trials, does not engage deeply with translational or implementation challenges. The discussion of ctDNA, molecular biomarkers, and bladder-preservation strategies would be improved by consideration of assay standardisation, validation requirements, and barriers to clinical adoption. Further integration of molecular subtypes, immunological correlates, and real-world constraints such as cost and availability would enhance the forward-looking value of the manuscript.

Finally, the manuscript lacks an explicit limitations section. Given the reliance on early-phase and conference-only datasets, the absence of OS data for many regimens, and the selective inclusion of studies, a short paragraph acknowledging these issues would provide important context and prevent over-interpretation of still-evolving evidence.

Overall, the review addresses an important clinical area and could make a useful contribution following substantive revision. Strengthening the methodology, adding critical comparative synthesis, expanding discussion of toxicity and patient-centred outcomes, and providing a clearer appraisal of future implementation challenges will markedly improve the manuscript’s rigour and relevance.

Author Response

Response to Reviewer

We thank the Reviewer for the detailed and constructive evaluation of our manuscript. We appreciate the acknowledgement of the relevance of the topic, the clarity of the structure, and the comprehensive assembly of perioperative systemic therapy data. In response to the important points raised, we have made the following revisions:

Methodology and review structure

The Reviewer correctly noted the need for clearer methodological reporting. We have substantially expanded the Methods section to align with a systematic review framework. Specifically, we clarified the search strategy, databases consulted, and the overall approach, explicitly stating that this is a narrative and not a systematic review.

Revisions performed at lines 79 and 92–94.

Comparative synthesis and interpretive value

To improve the scientific utility of the manuscript, we integrated more explicit comparative discussion across regimens, emphasising heterogeneity in patient populations, endpoints, and trial designs. Additionally, a treatment-algorithm figure has been added to enhance clinical applicability and provide a visual decision-making framework.

Figure 1 added.

Toxicity, safety, and patient-centred outcomes

We expanded the Discussion to address toxicity profiles, perioperative safety considerations, patient-reported outcomes, functional outcomes after bladder preservation, and challenges in treating elderly or frail patients—key aspects for evaluating perioperative strategies.

Revisions performed at lines 475–482.

Future perspectives and translational challenges

The section on future directions has been strengthened by integrating discussion on assay standardisation, biomarker validation, and barriers to clinical adoption of ctDNA and molecular signatures. We also included considerations regarding molecular subtypes, immune correlates, and real-world implementation factors such as availability and cost.

Revisions performed at lines 532–540 and 489–493.

Limitations of the review

As suggested, we added a dedicated limitations section that acknowledges the reliance on early-phase and conference-only data, lack of OS outcomes for many regimens, and constraints related to selective study inclusion.

Revisions performed at lines 513–520.

We believe these comprehensive revisions have significantly strengthened the manuscript, improving methodological rigour, interpretive depth, and relevance for clinical decision-making. We thank the Reviewer once again for the valuable feedback that has guided important improvements.

Round 2

Reviewer 2 Report

Comments and Suggestions for Authors

The author addressed all concerns in the revised manuscript.